# Localized JNK signaling regulates organ size during development

Helen Rankin Willsey[1], Xiaoyan Zheng[2†], José Carlos Pastor-Pareja[1‡], A Jeremy Willsey[3], Philip A Beachy[2], Tian Xu[1,4*]

[1]Department of Genetics, Howard Hughes Medical Institute, Yale University School of Medicine, New Haven, United States; [2]Departments of Biochemistry and Developmental Biology, Institute for Stem Cell Biology and Regenerative Medicine, Howard Hughes Medical Institute, Stanford University School of Medicine, Stanford, United States; [3]Department of Psychiatry, University of California, San Francisco, San Francisco, United States; [4]State Key Laboratory of Genetic Engineering and National Center for International Research, Fudan-Yale Biomedical Research Center, Institute of Developmental Biology and Molecular Medicine, School of Life Sciences, Fudan University, Shanghai, China

*For correspondence: tian.xu@ yale.edu

Present address: [†]Department of Anatomy and Regenerative Biology, School of Medicine and Health Sciences, The George Washington University, Washington, United States; [‡]School of Life Sciences, Tsinghua University, Beijing, China

Competing interests: The authors declare that no competing interests exist.

**Abstract** A fundamental question of biology is what determines organ size. Despite demonstrations that factors within organs determine their sizes, intrinsic size control mechanisms remain elusive. Here we show that *Drosophila* wing size is regulated by JNK signaling during development. JNK is active in a stripe along the center of developing wings, and modulating JNK signaling within this stripe changes organ size. This JNK stripe influences proliferation in a non-canonical, Jun-independent manner by inhibiting the Hippo pathway. Localized JNK activity is established by Hedgehog signaling, where Ci elevates *dTRAF1* expression. As the *dTRAF1* homolog, *TRAF4*, is amplified in numerous cancers, these findings provide a new mechanism for how the Hedgehog pathway could contribute to tumorigenesis, and, more importantly, provides a new strategy for cancer therapies. Finally, modulation of JNK signaling centers in developing antennae and legs changes their sizes, suggesting a more generalizable role for JNK signaling in developmental organ size control.

## Introduction

Within a species, organ size is remarkably reproducible. While extrinsic factors like hormones are required for growth, classic transplantation experiments indicate that intrinsic factors within organs determine size (*Bryant and Simpson, 1984*). For example, embryonic limb buds transplanted from a large species of salamander onto a small species grow to the size characteristic of the donor (*Twitty and Schwind, 1931*). Similar findings have been made in quail and chick limbs (*Iten and Murphy, 1980*; *Wolpert, 1978*), rat hearts and kidneys (*Dittmer et al., 1974*; *Silber, 1976*), and mouse thymuses (*Metcalf, 1963*). Consistently, developing *Drosophila* wings transplanted into adult abdomens grow to the proper size, indicating that the information determining size is located within the developing organ (*García-Bellido, 1965*). Indeed, the *Drosophila* wing is a classic model system for studying organ size, as its size is highly replicable (*García-Bellido and Merriam, 1971*; *García-Bellido, 1965*), and all adult precursor cells are located within the pouch region of the developing larval imaginal disc (*García-Bellido et al., 1973*) (*Figure 1A*, grey). Despite extensive work, the molecular mechanisms underlying intrinsic organ size control remain unclear (*Vogel, 2013*). While morphogens direct both patterning and growth of developing organs (*Tabata and Takei, 2004*), a

**eLife digest** A key challenge in biology is to understand what determines size. As an animal grows, signals are produced that control the size of its organs. Many of the signaling pathways that regulate size during normal animal development also contribute to the formation of tumors. Therefore, it is important to find out exactly how the signaling molecules that regulate size are linked to those that regulate tumor growth.

A protein called JNK activates a signaling pathway that triggers tumor growth. JNK signaling also stimulates cells to multiply in tissues that need repair, but it is not known whether it also regulates the size of organs during animal development. Here, Willsey et al. investigate whether JNK is active in the developing wings of fruit flies, which are commonly used as models of animal development.

The experiments show that JNK is active in a stripe across the developing wing and is required for the wing to grow to its proper size. A master signal protein called Hedgehog is responsible for establishing this stripe of JNK activity. Unexpectedly, rather than acting through its usual signaling pathway, JNK activates the Hippo pathway in the wing to control organ size during development.

Willsey et al.'s findings highlight potential new targets for cancer therapies. A future challenge will be to find out whether small patches of JNK signaling are found in the developing organs of other animals, and whether they can help explain how size changes between species.

link between patterning molecules and growth control pathways has not been established (*Schwank et al., 2011*).

The Jun N-terminal Kinase (JNK) pathway promotes proliferation during regeneration and tumor growth (*Bosch et al., 2005*; *Igaki et al., 2006*; *Ryoo et al., 2004*; *Srivastava et al., 2007*; *Wu et al., 2010*). In fact, JNK-induced proliferation is often non-autonomous (*Enomoto and Igaki, 2012*; *Pastor-Pareja et al., 2008*; *Ryoo et al., 2004*; *Sun and Irvine, 2011*; *Wu et al., 2010*). Basket (Bsk) is the singular *Drosophila* JNK and is activated by phosphorylation by the JNKK Hemipterous (Hep) (*Glise et al., 1995*; *Stronach, 2005*). Canonical JNK signaling acts through the transcription co-factor Jun, which regulates migration and apoptosis (*Stronach, 2005*). Although the role of JNK in activating Yorkie signaling and growth during regeneration and tumorigenesis is clear (*Enomoto and Igaki, 2012*; *Sun and Irvine, 2011*; *Sun and Irvine 2013*), it is not known to regulate proliferation and growth during developmental size control.

Here we show that localized JNK activity in the developing wing is established by Hedgehog (Hh) signaling and controls wing size through a non-canonical, Jun-independent signaling mechanism that inhibits the Hippo pathway.

## Results and discussion

### JNK is active in the developing *Drosophila* wing pouch

Two independently generated antibodies that recognize the phosphorylated, active form of JNK (pJNK) specifically label a stripe in the pouch of developing wildtype third instar wing discs (*Figure 1B–C* and *Figure 1—figure supplement 1G–H*). Importantly, localized pJNK staining is not detected in hemizygous *JNKK* mutant discs (*Figure 1D–E*; *hep$^{r75}$/Y*), in clones of *JNKK* mutant cells within the stripe (*Figure 1F*; *hep$^{r75}$, FRT10/Ubi-GFP, FRT10;; MKRS, hs-FLP/+*), following over-expression of the JNK phosphatase *puckered* (*puc*) (*Figure 1—figure supplement 1I*; *ap-Gal4, UAS-puc*), or following RNAi-mediated knockdown of *bsk* using two independent, functionally validated RNAi lines (*Figure 1—figure supplement 1K–L*; *rn-Gal4, UAS-bsk$^{RNAi#1}$* or *ptc-Gal4, UAS-bsk$^{RNAi#2}$*; see Experimental Genotypes for full genotypes and conditions) (*Glise et al., 1995*; *MacDonald et al., 2013*; *Martín-Blanco et al., 1998*; *Pérez-Garijo et al., 2013*; *Weber et al., 2000*; *Xu and Rubin, 1993*).

The stripe of localized pJNK staining appeared to be adjacent to the anterior-posterior (A/P) compartment boundary, a location known to play a key role in organizing wing growth, and a site of active Hedgehog (Hh) signaling (*Basler and Struhl, 1994*; *Tabata and Kornberg, 1994*; *Zecca et al., 1995*). Indeed, pJNK co-localizes with the Hh target gene *patched* (*ptc*) (*Figure 1G*;

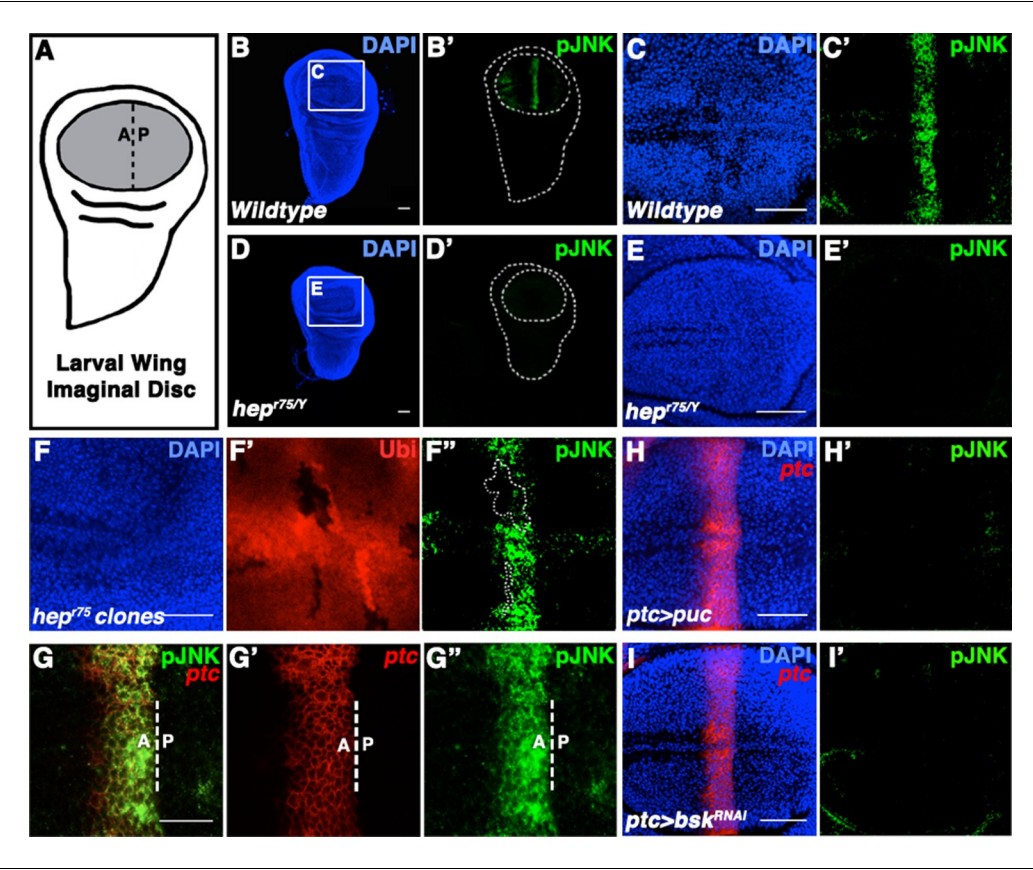

**Figure 1.** Localized JNK activity exists in the developing wing. (**A**) Schematic of wing precursor cells (grey) in the developing disc (A, anterior; P, posterior). (**B-F**) Antibody staining against active, phosphorylated JNK (pJNK, green; DAPI, blue) labels a stripe in wildtype (**B-C**) but not JNKK mutant (**D-E**, $hep^{r75/Y}$) third instar discs. Boxed region in (**B**) and (**D**) is magnified in (**C**) and (**E**), respectively. Weak pJNK signal is also detected along the dorsal/ventral boundary. pJNK stripe staining is lost in JNKK mutant clones (**F**, $hep^{r75}$, clone is negatively marked in **F'**). (**G-I**) pJNK localizes to the same cells in which *ptc* is expressed (**G**, *ptc>RFP*, red) along the A/P boundary, and is lost following JNK phosphatase expression (**H**, *ptc>puc*, RFP, red) or RNAi-mediated knockdown of *bsk* within the *ptc* domain (**I**, *ptc>bsk^{RNAi}*, RFP, red). Bar: 50 um (**B-F**, **H-I**) and 25 um (**G**). See also *Figure 1—figure supplement 1*.

The following figure supplement is available for figure 1:

**Figure supplement 1.** pJNK recognizes endogenous JNK activity in developing wing discs.

---

*ptc-Gal4, UAS-RFP*). Expression of the JNK phosphatase *puc* in these cells specifically abrogated pJNK staining (*Figure 1H*; *ptc-Gal4, UAS-puc*), as did RNAi-mediated knockdown of *bsk* (*Figure 1I* and *Figure 1—figure supplement 1L*; *ptc-Gal4, UAS-bsk^{RNA#i1or2}*). Together, these data indicate that the detected pJNK signal reflects endogenous JNK signaling activity in the *ptc* domain, a region of great importance to growth control. Indeed, while at earlier developmental stages pJNK staining is detected in all wing pouch cells (*Figure 1—figure supplement 1A*), the presence of a localized stripe of pJNK correlates with the time when the majority of wing disc growth occurs (1000 cells/disc at mid-L3 stage to 50,000 cells/disc at 20 hr after pupation, (*Garcia-Bellido, 2009*), so we hypothesize that localized pJNK plays a role in regulating growth.

## Localized JNK activity regulates global wing size

Inhibition of JNK signaling in the posterior compartment previously led to the conclusion that JNK does not play a role in wing development (*McEwen and Peifer, 2005*). The discovery of an anterior stripe of JNK activity spurred us to re-examine the issue. Since *bsk* null mutant animals are

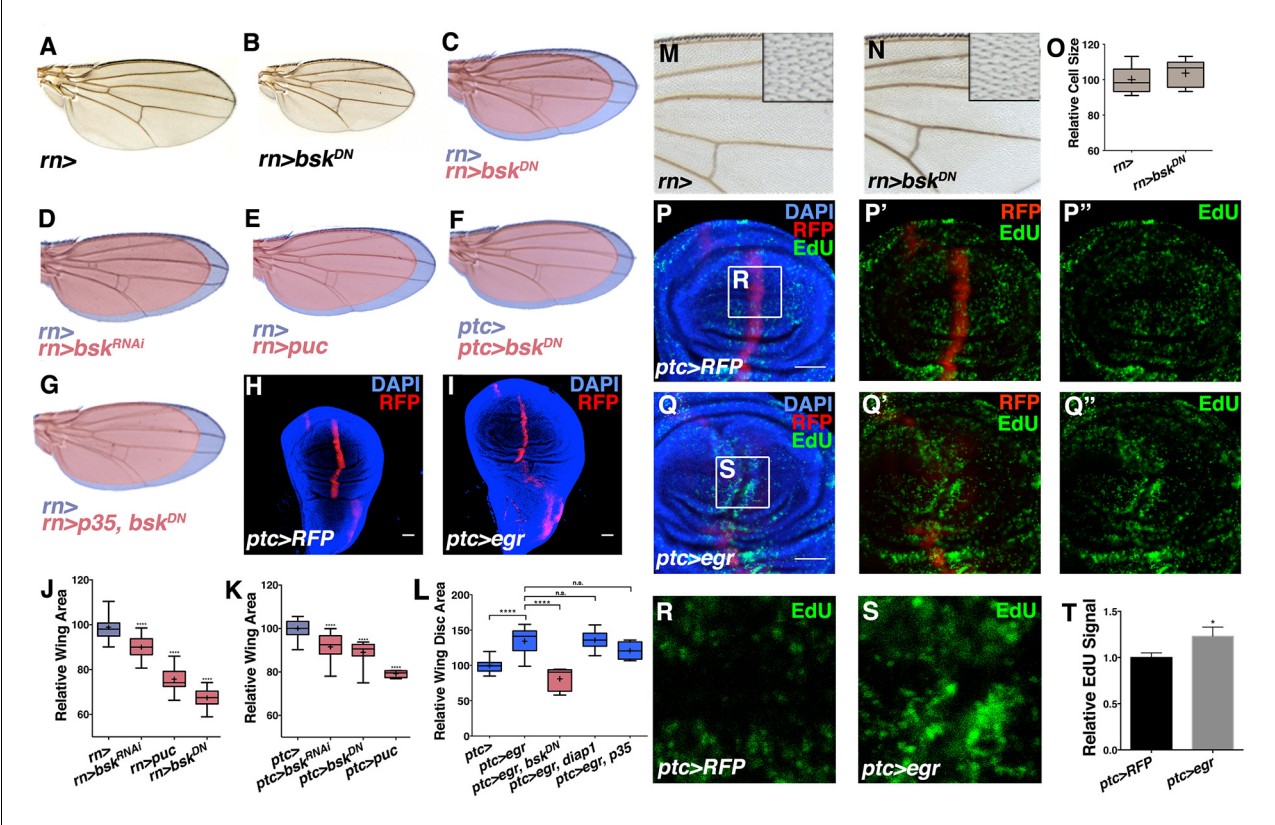

**Figure 2.** Modulation of localized JNK signaling changes wing size. Inhibition of JNK in all wing blade cells (B-E, J) or within the *ptc* domain (F, K) decreases adult wing size compared to controls (A, C-E, J, *rn>*) or (F, K, *ptc>*). Note that autonomous reduction between longitudinal veins 3 and 4 accounts for a small portion of the global reduction. Apoptosis inhibition does not rescue the small wing phenotype (red, G, *rn>p35, bsk^DN*). (H-I, L) Increased JNK signaling within the *ptc* domain following *eiger* expression causes an increase in disc size (I, *ptc>egr, RFP*, red; DAPI, blue) compared to controls (H, *ptc>RFP*, red). (L) This is increase is dependent on *bsk* (*ptc>egr, bsk^DN*) but not affected by *diap1* or *p35* expression (*ptc>egr, diap1* or *ptc>egr, p35*). Due to high pupal lethality, disc size was analyzed when animals reached the wandering third instar stage. (M-O) JNK inhibition does not affect cell size (N-O, *rn>bsk^DN*). (P-Q) Increased JNK signaling within the *ptc* domain causes an increase in proliferation (Q, *ptc>egr, RFP*, red; EdU, green) compared to controls (P, *ptc>RFP*, red; EdU, green). EdU of boxed region in (P) and (Q) is shown in (R) and (S), respectively. (T) Quantification of mean EdU signal in wing pouch regions between *ptc>RFP* and *ptc>egr* animals. Whiskers are SD. For box plots of area quantifications, whiskers represent maximum and minimum values (J-L, O). *-****=p<0.05–0.0001. n.s.= not significant. Bar: 50 um. See also *Figure 2—figure supplements 1–4*.

The following figure supplements are available for figure 2:

**Figure supplement 1.** JNK inhibition does not affect body size or cell death, but rather cell proliferation.

**Figure supplement 2.** Activating JNK signaling increases wing disc size independent of cell death or developmental timing.

**Figure supplement 3.** JNK inhibition does not affect Dpp or EGFR signaling.

**Figure supplement 4.** Inhibiting EGFR or Dpp signaling does not affect pJNK establishment.

embryonic lethal, we thus conditionally inhibited JNK signaling in three independent ways in the developing wing disc. JNK inhibition was achieved by RNAi-mediated knockdown of *bsk* (*bsk^RNAi#1or2*), by expression of JNK phosphatase (*puc*), or by expression of a dominant negative *bsk* (*bsk^DN*). These lines have been independently validated as JNK inhibitors (*MacDonald et al., 2013*; *Martín-Blanco et al., 1998*; *Perez-Garijo et al., 2013*; *Weber et al., 2000*). Inhibition of JNK in all wing blade cells (*rotund-Gal4, rn-Gal4*) or specifically in *ptc*-expressing cells (*ptc-Gal4*) resulted in smaller adult wings in all cases, up to 40% reduced in the strongest cases (*Figures 2A–F, 2J–K*, and *Figure 2—figure supplement 1D*). Importantly, expression of a control transgene (*UAS-GFP*) did

not affect wing size (*Figure 2—figure supplement 1B–C*; *ptc-Gal4, UAS-GFP*). This contribution of JNK signaling to size control is likely an underestimate, as the embryonic lethality of *bsk* mutations necessitates conditional, hypomorphic analysis. Nevertheless, hypomorphic *hep^{r75}/Y* animals, while pupal lethal, also have smaller wing discs (*Figure 2—figure supplement 1G*), as do animals with reduced JNK signaling due to *bsk^{DN}* expression (*Figure 2—figure supplement 1H–I*; *ap-Gal4, UAS-bsk^{DN}*). Importantly, total body size is not affected by inhibiting JNK in the wing. Even for the smallest wings generated (*rn-Gal4, UAS-bsk^{DN}*), total animal body size is not altered (*Figure 2—figure supplement 1A,E*).

To test whether elevation of this signal can increase organ size, we expressed *eiger* (*egr*), a potent JNK activator (*Igaki et al., 2002*), within the *ptc* domain (*ptc-Gal4, UAS-egr*). Despite induction of cell death as previously reported (*Igaki et al., 2002*) and late larval lethality, we observed a dramatic increase in wing disc size without apparent duplications or changes in the shape of the disc (*Figures 2H–I and 2L*; *ptc-Gal4, UAS-egr*). While changes in organ size could be due to changing developmental time, wing discs with elevated JNK signaling were already larger than controls assayed at the same time point (*Figure 2—figure supplement 2A–C*; *ptc-Gal4* and *ptc-Gal4, UAS-egr*). Similarly, inhibition of JNK did not shorten developmental time (*Figure 2—figure supplement 1F*; *rn-Gal4, UAS-bsk^{DN}*). Thus, changes in organ size by modulating JNK activity do not directly result from altering developmental time. Finally, the observed increase in organ size is not due to induction of apoptosis, as expression of the pro-apoptotic gene *hid* does not increase organ size (*Figure 2—figure supplement 2D–F*). In contrast, it causes a decrease in wing size (*Figure 2—figure supplement 2D–F*). Furthermore, co-expression of *diap1* or *p35* did not significantly affect the growth effect of *egr* expression ($p > 0.05$; *Figure 2L* and *Figure 2—figure supplement 2H–I*; *ptc-Gal4, UAS-egr, UAS-diap1* and *ptc-Gal4, UAS-egr, UAS-p35*), while the effect was dependent on Bsk activity ($p < 0.05$; *Figure 2L* and *Figure 2—figure supplement 2G*; *ptc-Gal4, UAS-egr, UAS-bsk^{DN}*).

In stark contrast to known developmental morphogens, we did not observe any obvious defects in wing venation pattern following JNK inhibition (*Figure 2A–B*), suggesting that localized pJNK may control growth in a pattern formation-independent manner. Indeed, even a slight reduction in Dpp signaling results in dramatic wing vein patterning defects (*Figure 2—figure supplement 3K*). Second, inhibiting Dpp signaling causes a reduction in wing size along the A-P axis, while JNK inhibition causes a global reduction (*Figure 2—figure supplement 3J–L*). Furthermore, ectopic Dpp expression increases growth in the form of duplicated structures (*Zecca et al., 1995*), while increased JNK signaling results in a global increase in size (*Figure 2H–I*). Molecularly, we confirm that reducing Dpp signaling abolishes pSMAD staining, while quantitative data shows that inhibiting JNK signaling does not (*Figure 2—figure supplement 3D–I*). Furthermore, we also find that Dpp is not upstream of pJNK, as reduction in Dpp signaling does not affect pJNK (*Figure 2—figure supplement 4B*). Together, the molecular data are consistent with the phenotypic results indicating that pJNK and Dpp are separate programs in regulating growth. Consistent with our findings, during the revision of this manuscript, it has been suggested that Dpp does not play a primary role in later larval wing growth control (*Akiyama and Gibson, 2015*). Finally, we found that inhibition of JNK does not affect EGFR signaling (pERK) and that inhibition of EGFR does not affect the establishment of pJNK (*Figure 2—figure supplement 3A–C* and *4A*).

A difference in size could be due to changes in cell size and/or number. We examined wings with reduced size due to JNK inhibition and did not detect changes in cell size or apoptosis (*Figure 2M–O* and *Figure 2—figure supplement 1L–N*; *rn-Gal4, UAS-bsk^{DN}*), suggesting that pJNK controls organ size by regulating cell number. Consistently, the cell death inhibitor *p35* was unable to rescue the decreased size following JNK inhibition (*Figure 2G*; *rn-Gal4, UAS-p35, UAS-bsk^{DN}*). Indeed, inhibition of JNK signaling resulted in a decrease in proliferation (*Figure 2—figure supplement 1J–K*; *ap-Gal4, UAS-bsk^{DN}*), while elevation of JNK signaling in the *ptc* domain resulted in an increase in cell proliferation in the enlarged wing disc (*Figure 2P–T*; *ptc-Gal4, UAS-egr*). Importantly, this increased proliferation is not restricted to the *ptc* domain, consistent with previous reports that JNK can promote proliferation non-autonomously (*Enomoto and Igaki, 2012*; *Pastor-Pareja et al., 2008*; *Ryoo et al., 2004*; *Sun and Irvine, 2011*; *Wu et al., 2010*).

## Non-canonical JNK signaling regulates size

To determine the mechanism by which pJNK controls organ size, we first considered canonical JNK signaling through its target Jun (*Ip and Davis, 1998*). Interestingly, RNAi-mediated knockdown of

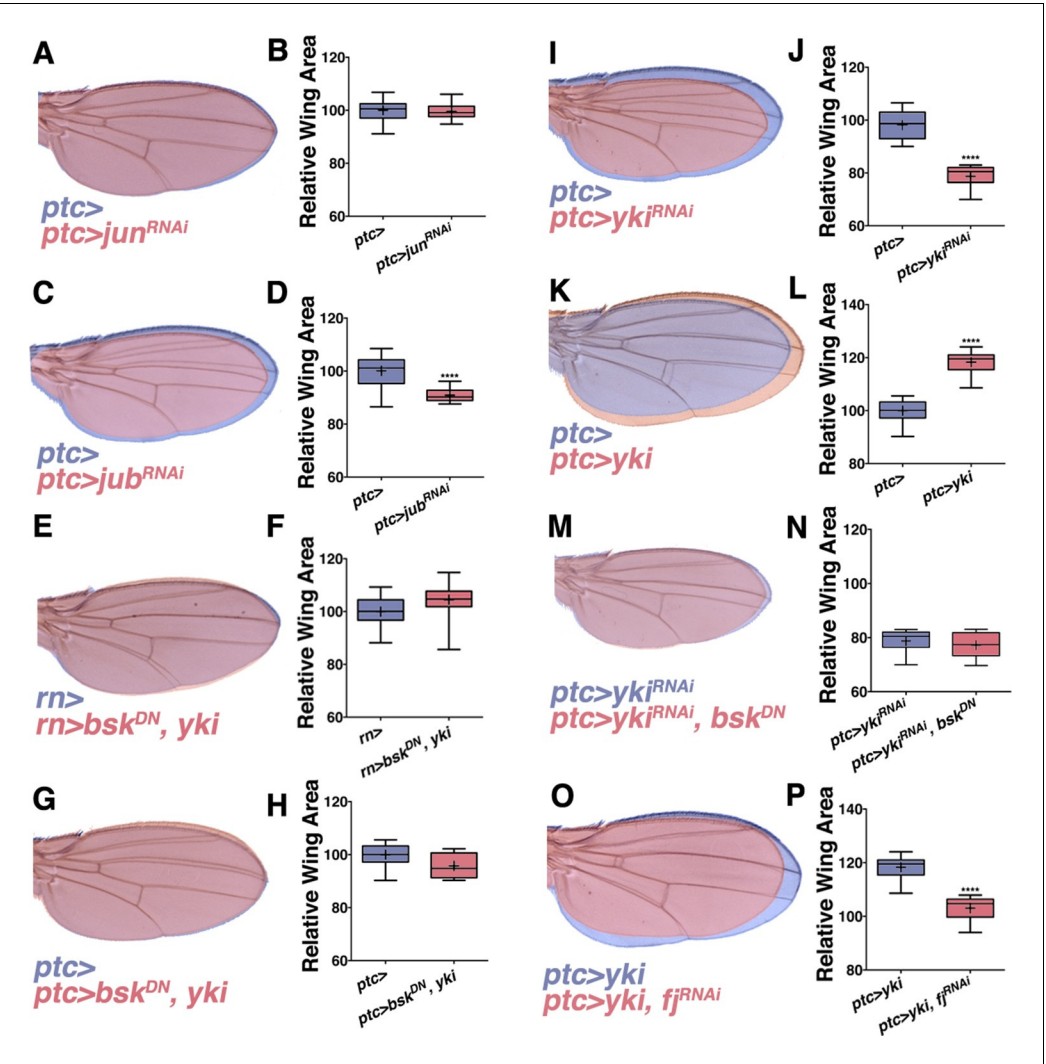

**Figure 3.** Non-canonical JNK signaling regulates wing size. RNAi-mediated knockdown of *Jun* within the *ptc* stripe does not change adult wing size (**A-B**, red, *ptc>jun^RNAi* compared to blue, *ptc>*). RNAi-mediated knockdown of *jub* does change global wing size (**C-D**, red, *ptc>jub^RNAi* compared to blue, *ptc>*). Expression of *yki* in all wing cells (**E-F**, red, *rn>yki, bsk^DN* compared to blue, *rn>*) or within the *ptc* stripe (**G-H**, red, *ptc>bsk^DN, yki* compared to blue, *ptc>*) rescues wing size following JNK inhibition. RNAi-mediated knockdown or overexpression of *yki* in *ptc* cells decreases or enlarges wing size, respectively (**I-J**, red, *ptc>yki^RNAi*, blue, *ptc>*, and K-L, red, *ptc>yki*, blue, *ptc>*). (**M-N**) Inhibition of JNK signaling does not enhance the phenotype of Yki inhibition alone (M, red, *ptc>bsk^DN, yki^RNAi*; blue, *ptc>yki^RNAi*). (**O-P**) RNAi-mediated knockdown of *fj* modifies the Yki growth phenotype (**O**, red, *ptc>yki, fj^RNAi*; blue, *ptc>yki*). For box plots, whiskers represent maximum and minimum values. ****=p<0.0001. See also *Figure 3—figure supplements 1–2*.

The following figure supplements are available for figure 3:

**Figure supplement 1.** *Jun* RNAi line validation and loss of *kayak* phenotypes.

**Figure supplement 2.** JNK interacts with Yki to cause global changes in wing size.

*jun* in *ptc* cells does not change wing size (*Figure 3A–B* and *Figure 3—figure supplement 1C–F*; *ptc-Gal4, UAS-jun^RNAi#1or2*; Both RNAi lines can effectively inhibit *jun* activity, *Figure 3—figure supplement 1A–B*), which is consistent with previous analysis of *jun* mutant clones in the wing disc (*Kockel et al., 1997*). Furthermore, in agreement with this, a reporter of canonical JNK signaling downstream of *jun* (*puc-lacZ* [*Ring and Martinez Arias, 1993*]) is not expressed in the pJNK stripe

(*Figure 1—figure supplement 1F*). Finally, knockdown of *fos* (*kayak, kay*) alone or with *jun^{RNAi}* did not affect wing size (*Figure 3—figure supplement 1G–H*; *rn-Gal4, UAS-kay^{RNiA#1or2}* and *rn-Gal4, UAS-jun^{RNAi#1}, UAS-kay^{RNiA#1or2}*). Together, these data indicate that canonical JNK signaling through *jun* does not function in the pJNK stripe to regulate wing size.

In search of such a non-canonical mechanism of JNK-mediated size control, we considered the Hippo pathway. JNK signaling regulates the Hippo pathway to induce autonomous and non-autonomous proliferation during tumorigenesis and regeneration via activation of the transcriptional regulator Yorkie (Yki) (*Bakal et al., 2008*; *Enomoto and Igaki, 2012*; *Sun and Irvine, 2011*). Recently it has been shown that JNK activates Yki via direct phosphorylation of Jub (*Sun and Irvine, 2013*). To test whether this link between JNK and Jub could account for the role of localized pJNK in organ size control during development, we performed RNAi-mediated knockdown of *jub* in the *ptc* stripe, and observed adults with smaller wings (*Figure 3C–D*; *ptc-Gal4, UAS-jub^{RNAi#1or2}*). Indeed, the effect of JNK loss on wing size can be partially suppressed in a heterozygous *lats* mutant background (*Figure 3—figure supplement 2C–D*; *rn-Gal4, UAS-bsk^{DN}, lats^{e26-1}/+*) and increasing downstream *yki* expression in all wing cells (*Figure 3E–F*; *rn-Gal4, UAS-yki, UAS-bsk^{DN}*) or just within the *ptc* domain (*Figure 3G–H*; *ptc-Gal4, UAS-yki, UAS-bsk^{DN}*) can rescue wing size following JNK inhibition. These results suggest that pJNK controls Yki activity autonomously within the *ptc* stripe, leading to a global change in cell proliferation. This hypothesis predicts that the Yki activity level within the *ptc* stripe influences overall wing size. Consistently, inhibition of JNK in the *ptc* stripe translates to homogeneous changes in anterior and posterior wing growth (*Figure 3—figure supplement 2A–B*). Similarly, overexpression or inhibition of Yki signaling in the *ptc* stripe also results in a global change in wing size (*Figure 3I–L* and *Figure 3—figure supplement 2A–B*; *ptc-Gal4, UAS-yki*; *ptc-Gal4, UAS-yki^{RNAi}*).

It is important to note that the *yki* expression line used is wild-type Yki, which is still affected by JNK signaling. For this reason, the epistasis experiment was also performed with activated Yki, which is independent of JNK signaling (*UAS-yki^{S111A,S168A,S250A.V5}*; (*Oh and Irvine, 2009*). Expression of this activated Yki in the *ptc* stripe caused very large tumors and lethality (data not shown). Importantly, inhibiting JNK in this context did not affect the formation of these tumors or the lethality (data not shown; *ptc-Gal4, UAS-yki^{S111A,S168A,S250A.V5}, UAS-bsk^{DN}*). Furthermore, inhibiting both JNK and Yki together does not enhance the phenotype of Yki inhibition alone (*Figure 3M–N* and *Figure 3—figure supplement 2E–F*; *ptc-Gal4, UAS-yki^{RNAi}, UAS-bsk^{DN}* and *ptc-Gal4, UAS-yki^{RNAi}, UAS-puc*), further supporting the idea that Yki is epistatic to JNK, instead of acting in parallel processes.

Mutants of the Yki downstream target *four-jointed* (*fj*) have small wings with normal patterning, and *fj* is known to propagate Hippo signaling and affect proliferation non-autonomously (*Ambegaonkar et al., 2012*; *Harvey and Tapon, 2007*; *Strutt et al., 2004*; *Villano and Katz, 1995*; *Willecke et al., 2008*). Although RNAi-mediated knockdown of *fj* in *ptc* cells does not cause an obvious change in wing size, it is sufficient to block the Yki-induced effect on increasing wing size (*Figure 3O–P* and *Figure 3—figure supplement 2G–H*; *ptc-Gal4, UAS-yki, UAS-fj^{RNAi}* and *ptc-Gal4, UAS-fj^{RNAi}*). However, overexpression of *fj* also reduces wing size, which makes it not possible to test for a simple epistatic relationship (*ptc-Gal4, UAS-fj*; *Figure 3—figure supplement 2I–J*). Overall, these data are consistent with the notion that localized pJNK regulates wing size not by Jun-dependent canonical JNK signaling, but rather by Jun-independent non-canonical JNK signaling involving the Hippo pathway.

## Hh sets up pJNK by elevating *dTRAF1* expression

While morphogens direct both patterning and growth of developing organs (*Tabata and Takei, 2004*), a link between patterning molecules and growth control pathways has not been established (*Schwank et al., 2011*). pJNK staining is coincident with *ptc* expression (*Figure 1G*), suggesting it could be established by Hh signaling. During development, posterior Hh protein travels across the A/P boundary, leading to activation of the transcription factor Cubitus interruptus (Ci) in the stripe of anterior cells (*Domínguez et al., 1996*; *Schwartz et al., 1995*). To test whether localized activation of JNK is a consequence of Hh signaling through Ci, we performed RNAi-mediated knockdown of *ci* and found that the pJNK stripe is eliminated (*Figure 4A–B*; *ptc-Gal4, UAS-ci^{RNAi#1or2}*). Consistently, adult wing size is globally reduced (*Figures 4D and 4G*). In contrast, we do not observe a change in pJNK stripe staining following RNAi-mediated knockdown of *dpp* or *EGFR* (*Figure 2—*

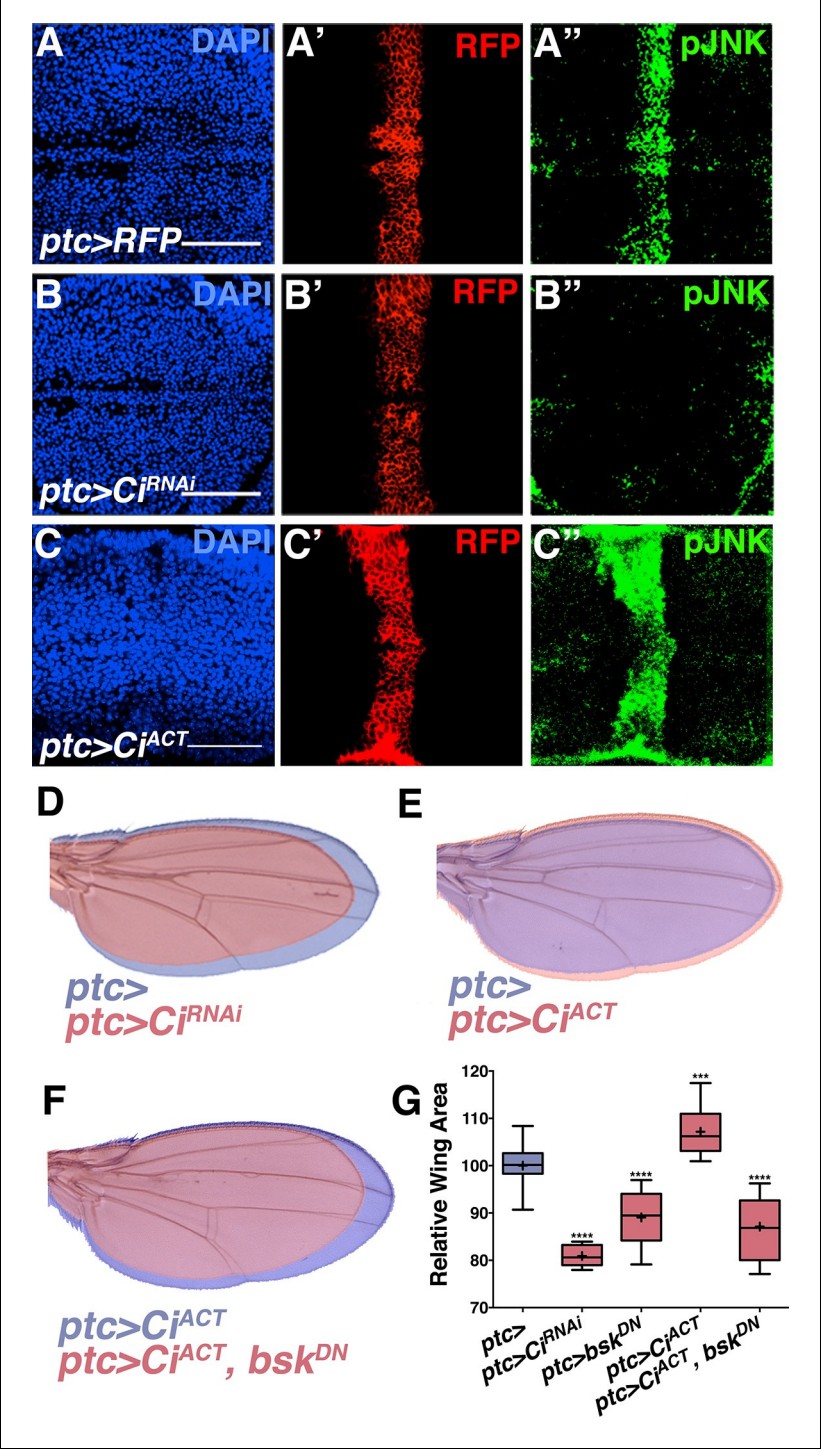

**Figure 4.** Hh signaling through Ci establishes localized pJNK. RNAi-mediated knockdown of *Ci* in *ptc* cells abrogates pJNK (green) staining (**A-B**, *ptc>Ci^RNAi*, *RFP* compared to *ptc>RFP*) and results in smaller adult wings (**D**, red, *ptc>Ci^RNAi* compared to blue, *ptc>*). Expression of activated *Ci* in the *ptc* domain leads to increased pJNK staining (green) (**C**, *ptc>Ci^ACT*, *RFP*) and a larger wing (**E**, red, *ptc>Ci^ACT* compared to blue, *ptc>*). Inhibition of JNK signaling in these cells blocks the effect of activated Ci (red, F, *ptc>Ci^ACT*, *bsk^DN*). For the box plot (**G**), whiskers represent maximum and minimum values. ***-****=p<0.001–0.0001. Bar: 50 um.

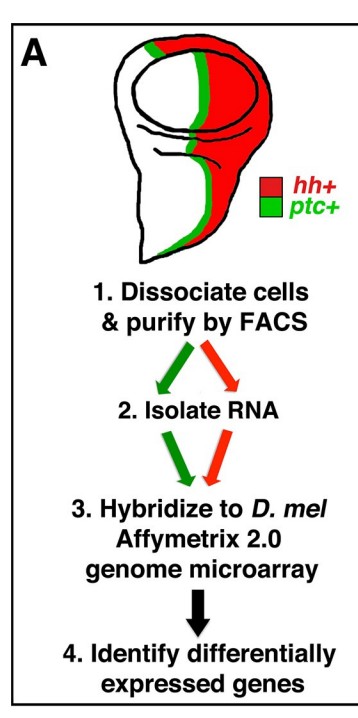

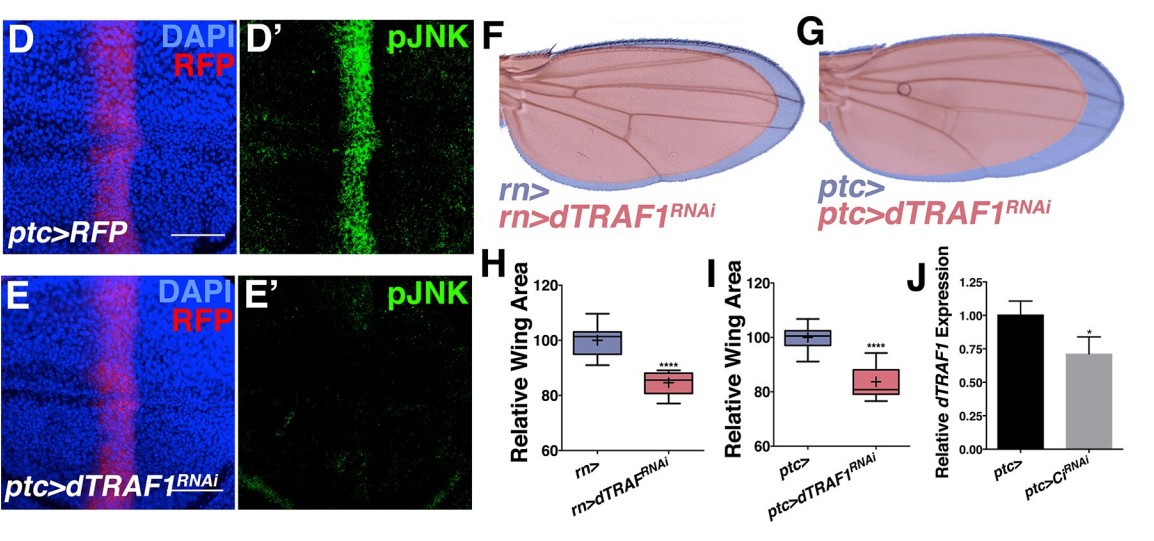

**B** Hh Pathway Genes

| Gene | Mean *ptc* Domain Log2 Expression | Mean *Hh* Domain Log2 Expression | Fold Change |
|------|-----------------------------------|----------------------------------|-------------|
| *en* | 10.4 | 11.5 | 0.5 |
| *hh* | 7.7 | 10.6 | 0.1 |
| *ci/Gli* | 11.9 | 7.9 | 14.6 |
| *dpp/TGF-β* | 12.2 | 8.5 | 14.3 |
| *ptc* | 7.0 | 5.4 | 2.8 |
| *kn* | 10.1 | 8.6 | 2.6 |

**C** JNK Pathway Genes

| Gene | Mean *ptc* Domain Log2 Expression | Mean *Hh* Domain Log2 Expression | Fold Change |
|------|-----------------------------------|----------------------------------|-------------|
| *slpr/JNKKK* | 10.5 | 10.3 | 1.1 |
| *MKK4/JNKK* | 10.7 | 10.6 | 1.1 |
| *dTRAF1/TRAF4* | 9.0 | 6.6 | 5.1 |
| *msn/JNKKKK* | 10.1 | 9.7 | 1.2 |
| *dTAK1/JNKKK* | 9.8 | 9.7 | 1.0 |
| *hep/JNKK* | 8.9 | 8.9 | 0.9 |
| *bsk/JNK* | 11.0 | 11.1 | 1.0 |
| *Jra/Jun* | 10.2 | 10.5 | 0.8 |
| *kay/Fos* | 11.1 | 11.4 | 0.8 |
| *puc* | 10.9 | 10.8 | 1.1 |
| *rpr* | 8.9 | 9.5 | 0.7 |

**Figure 5.** Hedgehog signaling establishes pJNK by elevating *dTRAF1* expression. (**A**) *ptc* cells (green, *ptc+*) and posterior cells (red, *hh+*) from third instar wing discs were dissociated and sorted by FACS. RNA was isolated and hybridized to microarrays. Differentially expressed genes were identified. (**B**) Hedgehog pathway genes known to be differentially expressed are identified. Genes upregulated in *ptc* cells (*ptc+*) compared to posterior (*hh+*) cells are highlighted in green and downregulated in red. Genes with log$_2$ normalized expression ≥6.5 are considered expressed. (**C**) JNK pathway gene *dTRAF1* is >5-fold upregulated in *ptc* cells. (**D-I**) RNAi-mediated knockdown of *dTRAF1* eliminates pJNK (green) staining (**E**, *ptc>dTRAF$^{RNAi\#1}$*, RFP, red) and leads to smaller adult wings (**F-I**, *rn>dTRAF$^{RNAi\#1}$* or *ptc>dTRAF$^{RNAi\#1}$*). (**J**) Ci inhibition causes a ~30% decrease in *dTRAF1* expression in 3$^{rd}$ instar wing discs, relative to endogenous control *Rp49*. Whiskers are SD. For box plots, whiskers are maximum and minimum values (**H-I**). *-****=p<0.05–0.0001. Bar: 50 um. See also *Figure 5—figure supplement 1–2*.

The following figure supplements are available for figure 5:

**Figure supplement 1.** Transcriptional profiling quality control and additional dTRAF1 validation.
**Figure supplement 2.** Inhibiting *dTRAF1* can modify an activated Ci phenotype.

figure supplement 4A–B). Expression of non-processable Ci leads to increased Hh signaling (*Price and Kalderon, 1999*). Expression of this active Ci in *ptc* cells leads to an increase in pJNK signal and larger, well-patterned adult wings (*Figures 4C,E*, and 4G; *ptc-Gal4, UAS-Ci^ACT*). The modest size increase shown for *ptc>Ci^ACT* is likely due to the fact that higher expression of this transgene (at 25°C) leads to such large wings that pupae cannot emerge from their cases. For measuring wing size, this experiment was performed at a lower temperature (20°C, see Experimental Genotypes) so that the animals were still viable. Furthermore, inhibition of JNK in wings expressing active Ci blocks Ci's effects, and resulting wings are similar in size to JNK inhibition alone (*Figure 4F–G*; *ptc-Gal4, UAS-Ci^ACT, UAS-bsk^DN*). Together, these data indicate that Hh signaling through Ci is responsible for establishing the pJNK stripe.

To determine the mechanism by which Ci activates the JNK pathway, we compared transcriptional profiles of posterior (red, *hh+*) and *ptc* domain cells (green, *ptc+*) isolated by FACS from third instar wing discs (*Figure 5A*; Materials and methods). Of the total 12,676 unique genes represented on the microarray, 50.4% (6,397) are expressed in *ptc* domain cells, posterior cells, or both ($\log_2$ normalized expression $\geq 6.5$; *Figure 5—figure supplement 1A–D*; *Supplementary file 1*; Materials and methods). We thresholded on a false discovery rate <0.01 and fold change $\geq 1.5$ and found that 5.7% (363) of expressed genes were upregulated in *ptc* cells and 3.8% (242) were downregulated (*Figure 5—figure supplement 1D*; *Supplementary file 2*; Materials and methods). Hh pathway genes known to be differentially expressed are identified (*Figure 5B*). We next asked whether any JNK pathway genes are differentially expressed and found that *dTRAF1* expression is more than five-fold increased in *ptc* cells (*Figure 5C*), while other JNK pathway members are not differentially expressed (*Figure 5C*; *Supplementary file 1*; *Supplementary file 2*).

*dTRAF1* is expressed along the A/P boundary (*Preiss et al., 2001*) and ectopic expression of *dTRAF1* activates JNK signaling (*Cha et al., 2003*). Thus, positive regulation of *dTRAF1* expression by Ci could establish a stripe of pJNK that regulates wing size. Indeed, we identified Ci binding motifs in the *dTRAF1* gene (*Figure 5—figure supplement 1H*), and a previous large-scale ChIP study confirms a Ci binding site within the *dTRAF1* gene (Chr2L: 4367100- 4371393; [*Biehs et al., 2010*]). Consistently, a reduction in *Ci* led to a 29% reduction in *dTRAF1* expression in wing discs (*Figure 5J*; *ptc-Gal4, UAS-Ci^RNAi*). Given that the reduction of *dTRAF1* expression in the *ptc* stripe is buffered by Hh-independent *dTRAF1* expression elsewhere in the disc (*Preiss et al., 2001*), this 29% reduction is significant. Furthermore, inhibition of *dTRAF1* by RNAi knockdown abolished pJNK staining (*Figure 5D–E* and *Figure 5—figure supplement 1E*; *ptc-Gal4, UAS-dTRAF1^RNAi#1or2*). Finally, these animals have smaller wings without obvious pattern defects (*Figure 5F–I* and *Figure 5—figure supplement 1F–G*). Conversely, overexpression of *dTRAF1* causes embryonic lethality (*ptc-Gal4, UAS-dTRAF1*), making it not possible to attempt to rescue a *dTRAF1* overexpression wing phenotype by knockdown of *bsk*. Nevertheless, it has been shown that dTRAF1 function in the eye is Bsk-dependent (*Cha et al., 2003*). Finally, inhibition of *dTRAF1* modulates the phenotype of activated Ci signaling (*ptc-Gal4, UAS-dTRAF1^RNAi, UAS-Ci^ACT*; *Figure 5—figure supplement 2*). Together, these data reveal that the pJNK stripe in the developing wing is established by Hh signaling through Ci-mediated induction of *dTRAF1* expression.

## Localized pJNK controls antenna and leg size

Finally, we detected localized centers of pJNK activity during the development of other imaginal discs including the eye/antenna and leg (*Figures 6A and 6G*). Inhibition of localized JNK signaling during development caused a decrease in adult antenna size (*Figures 6B–C and 6F*; *dll-Gal4, UAS-bsk^DN*) and leg size (*Figures 6H–I and 6L*; *dll-Gal4, UAS-bsk^DN*). Conversely, increasing JNK signaling during development resulted in pupal lethality; nevertheless, overall sizes of antenna and leg discs were increased (*Figures 6D–E and 6J–K*; *dll-Gal4, UAS-egr*). Together, these data indicate that localized JNK signaling regulates size in other organs in addition to the wing, suggesting a more universal effect of JNK on size control.

Intrinsic mechanisms of organ size control have long been proposed and sought after (*Bryant and Simpson, 1984*; *Vogel, 2013*). Our study reveals that in developing *Drosophila* tissues, localized, organ-specific centers of JNK signaling contribute to organ size in an activity level-dependent manner. Such a size control mechanism is qualitatively distinct from developmental morphogen mechanisms, which affect both patterning and growth (*Zecca et al., 1995*). Aptly, this mechanism is still integrated in the overall framework of developmental regulation, as it is established in the wing

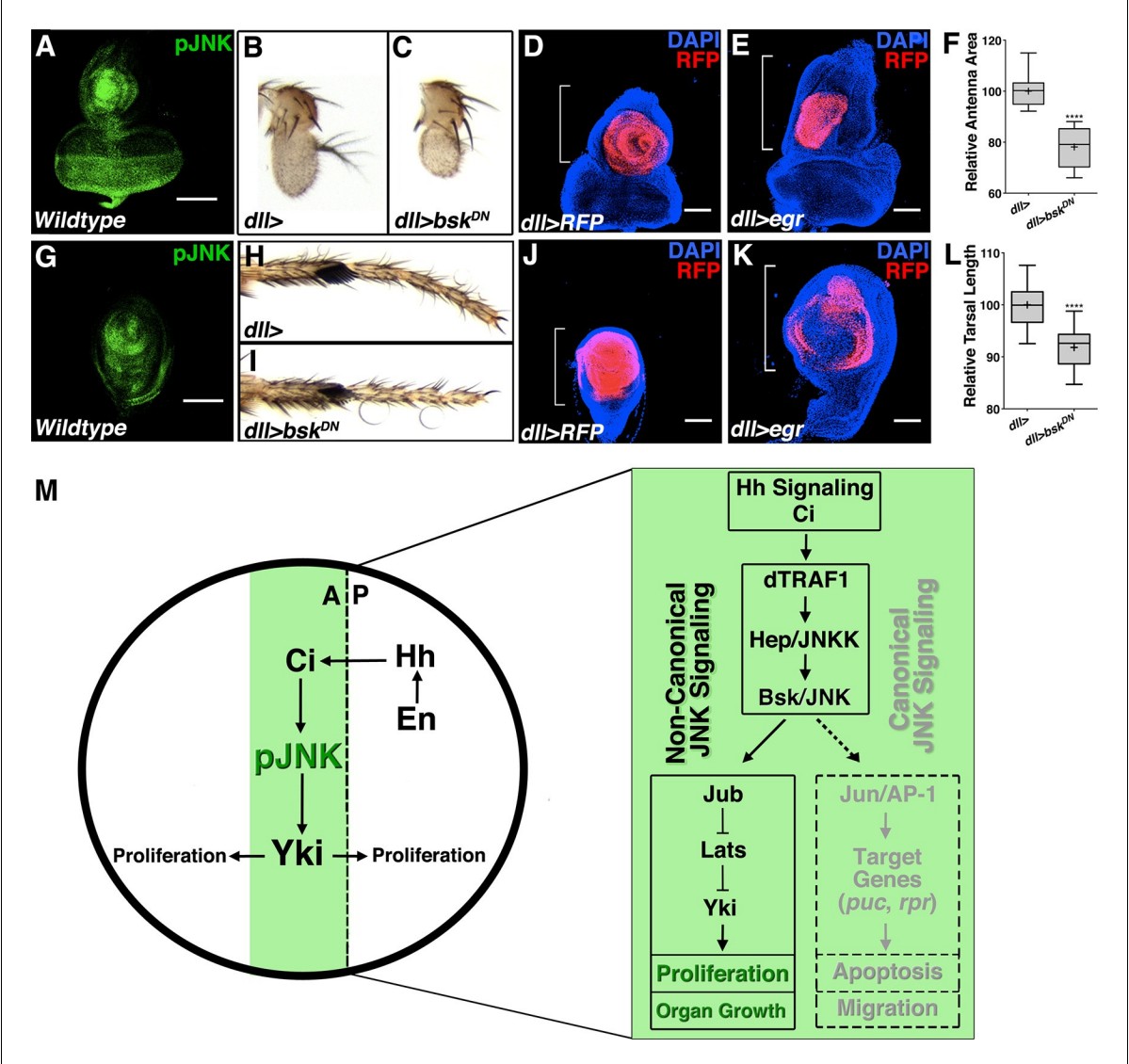

**Figure 6.** Modulation of localized JNK signaling within the developing antenna or leg changes organ size. pJNK (green) staining of wildtype antenna/eye (A) and leg (G) third instar discs. Inhibition of JNK in the developing antenna (B-C, F, *dll>bsk^DN*) or leg (H-I, L, *dll>bsk^DN*) leads to a smaller adult organ. Increased JNK activation within the antenna (D-E, *dll>egr*, *RFP*, red) or leg disc (J-K, *dll>egr*, *RFP*, red) causes an increase in disc size. (M) Model of how localized JNK signaling regulates wing size during development. Engrailed (En) controls Hh signaling, leading to a stripe of active Ci along the A/P boundary. Ci increases transcription of *dTRAF1*, activating JNK (pJNK, green). JNK acts in a non-canonical, Jun-independent manner to regulate Yki or Yki-dependent signaling. As the human *dTRAF1* homolog, *TRAF4*, and Hippo components are amplified in numerous cancers, these findings provide a new mechanism for how the Hh pathway could contribute to tumorigenesis (*Camilleri-Broët et al., 2006*; *Harvey et al., 2013*). For box plots, whiskers represent maximum and minimum values (F, L). ****=p<0.0001. Bar: 100 um

by the Hh pathway (*Figure 6M*). Our data indicate that localized JNK signaling is activated by Ci-mediated induction of *dTRAF1* expression. Furthermore, we discovered that it is not canonical Jun-dependent JNK signaling, but rather non-canonical JNK signaling that regulates size, possibly through Jub-dependent regulation of Yki signaling, as described for regeneration (*Sun & Irvine, 2013*) (*Figure 6M*). As the human *dTRAF1* homolog, *TRAF4*, and Hippo components are amplified in numerous cancers (*Camilleri-Broët et al., 2006*; *Harvey et al., 2013*), these findings provide a new mechanism for how the Hh pathway could contribute to tumorigenesis. More importantly, these findings offer a new strategy for potential cancer therapies, as reactivating Jun in Hh-driven tumors could lead tumor cells towards an apoptotic fate.

## Materials and methods

### *Drosophila* stocks and husbandry

Fly crosses were maintained at 25°C on standard cornmeal-molasses media unless otherwise indicated (see Experimental Genotypes). When possible, crosses were established so that every experimental animal had an in-vial *Gal4* alone control. For experiments that necessitated precise developmental staging, 2 hr egg lays were conducted on apple juice agar plates with yeast paste. For all other experiments, females were allowed to lay eggs on standard media for 24 hr, after which they were removed and progeny were considered as 12 +/- 12 hr after egg lay. The following stocks were utilized: (1) *Canton-S* (02) *y, hep^{r75}, FRT10.1/FM7iGFP* (*Glise et al., 1995*) (2) *Ubi-GFP, FRT10.1;; hs-FLP, MKRS/TM6B* (3) *UAS-puc* (III) (*Martín-Blanco et al., 1998*) (4) *w; ap-GAL4, UAS-src-RFP; Sb/TM6B* (5) *w; ptc-GAL4, UAS-src-RFP; Sb/TM6B* (6) *UAS-bsk^{RNAi}*(II and III) VDRC 34138 (*Perez-Garijo et al., 2013*) and BDSC 32977 (7) *w, UAS-bsk^{DN} (X)* (8) *w;; UAS-bsk^{DN}/TM6B* (9) *w;; rn-GAL4/TM6B* (10) *y, UAS-p35; Adv/CyO; Sb/TM6B* (11) *w; Sp/CyO; UAS-egr/MKRS* (12) *UAS-diap1* (III) BDSC 6657 (13) *UAS-bsk^{AY}* (II) BDSC: 6407 (14) *UAS-Ci^{RNAi}*(II and III) BDSC 31236 and 31236 (15) *UAS-Ci5m/TK-GFP* ("*UAS-Ci^{ACT}*") (*Price and Kalderon, 1999*) (16) *puc^{E69}/TM6B* ("*puc-lacZ*") (*Ring and Martinez Arias, 1993*) (17) *UAS-dTRAF1^{RNAi}*(X and III) VDRC 21213 and 21214 (18) *UAS-jun^{RNAi}* (III) BDSC 31595 and VDRC 10835 (19) *UAS-kay^{RNAi#1}* (III) BDSC 33379 and 31322 (20) *UAS-jub^{RNAi}*(III and II) BDSC 32923 and 41938 (21) *y,w;; lates^{e26-1}/TM6B* (22) *yw; UAS-yki.GFP; Sb/TM6B* BDSC 28815 (*Oh and Irvine, 2008*) (23) *UAS-yki^{RNAi}/TM3* BDSC 31965 (24) *UAS-fj^{RNAi}/TM3* BDSC 28009 (25) *UAS-fj.V5* (III) BDSC 44252 (26) *w; dll-Gal4, UAS-src.RFP/CyO* (27) *UAS-dpp^{RNAi}*(III) BDSC 25782 (28) *UAS-EGFR^{RNAi}*(III) BDSC 25781 (29) *UAS-yki^{S111A.S168A.S250A.V5}* (III) BDSC 28817

### Imaginal disc staining

Antibody staining was performed according to standard procedures for imaginal discs. The following antibodies were used: rabbit PhosphoDetect™ anti-SAPK/JNK (pThr^{183}, pTyr^{185}) (1:100, Calbiochem, immunogenic sequence is 100% identical to *D. melanogaster bsk/JNK*), rabbit anti-ACTIVE® JNK (1:100, Promega, immunogenic sequence is 100% identical to *D. melanogaster bsk/JNK*), rabbit anti-cleaved-caspase 3 (1:250, Cell Signaling), mouse anti-betagalactosidase (1:500, Sigma), rabbit anti-pERK (1:75, Cell Signaling), rabbit anti-pSMAD (1:75, Cell Signaling), rabbit anti-phosphorylated histone 3 (1:250, Cell Signaling), goat Alexa-488-conjugated anti-rabbit IgG (1:250, Invitrogen), goat Alexa-488-conjugated anti-mouse IgG (1:250, Invitrogen), goat Alexa-555-conjugated anti-rabbit IgG (1:250, Invitrogen). EdU staining was performed according to established protocol (*Gouge and Christensen, 2010*) using the Click-iT EdU cell proliferation assay kit (Invitrogen), Grace's Media (Invitrogen) and a 10 min EdU incubation.

### Imaginal disc imaging

Imaginal discs to be imaged by confocal microscopy were mounted in Vectashield mounting media with DAPI (Vector Labs). Confocal images were taken with a Zeiss LSM510 Meta confocal microscope or a Leica TCS SP8 STEAD 3X confocal microscope with 405nm, 488nm, 561nm, and 633nm lasers. Both microscopes gave similar results. Measurements of disc size were performed from images of at least fifteen discs using NIH Image-J software.

### Western blot analysis

Whole *Canton-S* and *hep^{r75/Y}* larvae were lysed in standard RIPA buffer with protease and phosphatase inhibitors. Proteins were separated by SDS-PAGE using a 4–15% acrylamide gel (BioRad), transferred for 1 hr at 4°C, and probed with primary antibodies: rabbit anti-pJNK (Calbiochem, 1:1000) and mouse anti-alpha tubulin (Sigma, 1:4000). HRP-conjugated secondary antibodies (anti-rabbit and anti-mouse) were used at 1:5000. ECL (Pierce) was used for detection with film.

### Adult organ imaging

Adult wings, legs, or antenna were dissected in 70% ethanol, mounted in Permount mounting media (Fisher Scientific), and imaged with a Leica DFC300FX camera on a Leica MZ FLIII stereomicroscope. Measurements of wing size were performed from images of twenty to sixty female flies using NIH Image-J software. Wing images were false-colored and overlayed to scale using Adobe Photoshop

CS3 software. Cell size was measured by dividing the number of hairs (1 hair/cell) by a set area using Adobe Photoshop CS3 software. Mean EdU signal was measured in Adobe Photoshop CS3. Measurements of antenna or leg size were performed from images of at least twenty male flies for each genotype using NIH Image-J software.

## Statistical analysis

To determine whether differences in area were statistically significant, two-sided student's t-tests were performed using raw data values, matched for temperature and sex. Box plots were generated where whiskers represent maximum and minimum, a plus sign indicates the mean, a horizontal line within the box indicates the median, and the box represents the 25–75% quartile range. Both parametric and non-parametric analyses were performed, and p-values less than 0.05 were considered significant. Data are presented as relative to the mean of the matched *Gal4*-alone control.

## Gene expression profiling

For each of three biological replicates, 200 pairs of wing imaginal discs were dissected from third instar larvae of the genotypes *hh-Gal4; UAS-mCD8GFP* or *ptc-Gal4; UAS-mCD8GFP*. Discs were stored in Schneider's *Drosophila* Media (21720, Invitrogen) plus 10% FBS (10438, Invitrogen) on ice for less than two hours prior to cell dissociation. Discs were washed twice with 1 ml cell dissociation buffer (Sigma, C-1544). Elastase (Sigma, E-0258) was diluted to 0.4 mg/ml in fresh cell dissociation buffer once discs were ready. Discs were incubated for 20 min at room temperature in 0.4 mg/ml elastase with stirring by a magnetic micro stirring bar. Undissociated tissue was spun out, cell viability was measured using the Beckman Vi-CELL Cell Viability Analyzer (>80%), and cells were immediately isolated using the BD FACSAria II system within the Stanford FACS facility. Dead cells labeled with propidium iodide (P3566, Invitrogen) were excluded during FACS, and purity of sorted cells was greater than 99% by post-sorting FACS analysis. Total RNA was extracted from sorted cells (RNeasy, Qiagen), quality was assessed with the Agilent Bioanalyzer 2100 (RIN > 7.0), and microarray analysis was performed in the Stanford Protein and Nucleic Acid Facility (Affymetrix *D. mel* Gene-Chip Genome 2.0 microarrays).

## Identification of differentially expressed genes

All analyses were conducted in R version 3.1.1 (2014-07-10). Expression values were determined using the *affy* package (*Gautier et al., 2004*), available from BioConductor (http://bioconductor.org). The automatically downloaded *Drosophila* 2.0 CDF environment was utilized. Probe level data from the *CEL* files were imported using the function *ReadAffy* and converted to expression values using the function *rma* with default settings. This method implements robust multi-array average (RMA) for background correction followed by quantile normalization. PM correction was not performed. Probe level expression values were combined into probe set expression measures using *medianpolish,* the standard summary method employed in RMA (*Irizarry et al., 2003*). Expression values are $\log_2$ transformed.

Post-normalization microarray quality assessment was conducted using the *arrayQualityMetrics* package (*Kauffmann et al., 2009*), available from BioConductor. Default settings were used, with *ptc* domain (*ptc+*) versus posterior (*hh+*) as the covariate in *intgroup*. Biological replicates cluster together in a dendrogram of inter-array difference, estimated as the mean absolute difference between the data of the arrays (*Figure 5—figure supplement 1A*), indicating that biological effects are stronger than any batch effects. Similarly, principle components analysis also separates biological replicates into two clusters (*Figure 5—figure supplement 1B*). Outliers were not detected by either of these methods.

Probe sets were mapped to genes using the *drosophila2.db* annotation package (version 3.0.0), available from BioConductor. 14,481 of 18,952 (76.4%) probe sets map to gene isoforms—12,676 (87.5%) of which correspond to unique genes (some genes are mapped by $\geq$1 probe set). In order to minimize technical artifacts, probe sets mapping to the same gene were not combined.

Based on the distribution observed in the density plot of normalized probe set expression values, probe sets (genes) with median $\log_2$ expression value $\geq$6.5 in at least one condition (*ptc+* and/or *hh +*) were considered to be expressed (*Figure 5—figure supplement 1C*). According to these criteria, 7,228 of 18,952 probe sets (38.1%) are expressed. This corresponds to 6,854 of 14,481 gene

isoforms (47.3%), which corresponds to 6,397 of 12,676 unique genes (50.4%, *Figure 5—figure supplement 1D*, *Supplementary file 1*).

To identify probe sets (genes) differentially expressed between *ptc+* and posterior (*hh+*) samples, we used the *samr* package, an R implementation of significance analysis of microarrays (*Tusher et al., 2001*). This package is available from CRAN (http://cran.r-project.org/). Only expressed probe sets mapping to genes (6,854) were considered in this analysis. Differentially expressed probe sets were identified with the function *SAM*, using a two class unpaired response type, the t-statistic as the test statistic, and a false discovery rate (FDR) threshold of 0.01. The maximum number of possible permutations (720) was used. To ensure these results are biologically meaningful, we further trimmed this list to probe sets with a minimum 1.5 fold change between *ptc+* and *hh+* cells. Based on these criteria, 624 of 6,854 probe sets (9.1%) are differentially expressed, with 376 (5.5%) upregulated in *ptc+* samples and 248 (3.6%) downregulated in *ptc+* samples (*Figure 5—figure supplement 1D*, *Supplementary file 2*). A gene was considered differentially expressed if any mapped probe set was differentially expressed. Therefore, of the 6,397 unique expressed genes, 604 (9.4%) are differentially expressed, 363 (5.7%) upregulated and 242 (3.8%) downregulated. One gene, *Tie*, was mapped by probe sets both up- and down-regulated. The quantile-quantile plot in *Figure 5—figure supplement 1D* was prepared using the *samr.plot* function.

## Real-time polymerase chain reaction

Total RNA was extracted from third instar wing discs from *ptc-Gal4* or *ptc-Gal4, UAS-Ci^RNAi* animals using a standard TriZol extraction. RNA was reverse transcribed using the iScript cDNA Synthesis Kit (Bio-Rad) according to manufacturer's instructions. *dTRAF1* expression was quantified relative to *Rp49* (*RpL32*- FlyBase, endogenous control) by real-time PCR performed in triplicate using the SYBR Green fast kit (Applied Biosystems) and an Applied Biosystems machine according to the manufacturer's instructions. The following primers were used: *dTRAF1*, 5'-GCACTCCATCACCTTCACAC-3' and 5'-TAGCTGATCTGGTTCGTTGG-3'; *Rp49*, 5'-GGCCCAAGATCGTGAAGAAG-3' and 5'-ATTTG TGCGACAGCTTAGCATATC-3'.

## Transcription factor binding site analysis

The *Drosophila* Ci positional weight matrix from the BioBase TRANSFAC database was queried against the *Drosophila melanogaster* genome with a p-value <0.0001 (chosen based on known Ci binding sites within *ptc*) using FIMO (MEME) and aligned back to the UCSC genome browser.

## Experimental genotypes

Crosses were maintained at 25°C unless otherwise indicated

*Figure 1*: (B-C) Canton-S (D-E) *y, hep^r75^, FRT10.1 /Y* (F) *y, hep^r75^, FRT10.1/Ubi-GFP, FRT10.1;; hs-FLP, MKRS/+* (G) *w/+; ptc-GAL4, UAS-src.RFP/+* (H) *w; ptc-GAL4, UAS-src.RFP; UAS-puc* 29°C (I) *w; ptc-GAL4, UAS-src.RFP/UAS-bsk^RNAi^* 29°C

*Figure 1—figure supplement 1*: (A-C, G-H) Canton-S, (D-F) *puc^E69^/+* (I) *w; ap-Gal4/+; UAS-puc/+* (J) *w; ptc-Gal4, UAS-src.RFP/+* (K) *w/yv, UAS-bsk^RNAi#1^/UAS-src.RFP; rn-Gal4/+* 29°C (L) *w/yv; ptc-Gal4, UAS-src.RFP/+; UAS-bsk^RNAi#2^/+*

*Figure 2*: (A) *w/+;; rn-Gal4/+* (B) *w/w, UAS-bsk^DN^;; rn-Gal4/UAS-bsk^DN^* (C) Blue: *w/+;; rn-Gal4/+* Red: *w/w, UAS-bsk^DN^;; rn-Gal4/UAS-bsk^DN^* (D) Blue: *w/+;; rn-Gal4/+* 29°C Red: *w; UAS-bsk^RNAi#1^/+; rn-Gal4/+* 29°C (E) Blue: *w/+;; rn-Gal4/+* 29°C Red: *w;; rn-Gal4, UAS-puc/UAS-puc* 29°C (F) Blue: *w/+; ptc-Gal4, UAS-src.RFP/+; Sb/+* Red: *w, UAS-bsk^DN^/w; ptc-GAL4, UAS-src.RFP/Sp; UAS-bsk^DN^/Sb* (G) *w, UAS-bsk^DN^/w, UAS-p35;; rn-GAL4/UAS-bsk^DN^* 29°C (H, P, R) *w/+; ptc-GAL4, UAS-src.RFP/+* (I, Q, S) *w; ptc-GAL4, UAS-src.RFP/+, UAS-egr/+* (M) *w/+;; rn-Gal4/+* (N) *w/w, UAS-bsk^DN^; Sp/+; rn-Gal4/UAS-bsk^DN^*

*Figure 2—figure supplement 1*: (A) Left: *w/+;; rn-Gal4/+* 25°C Right: *w/w, UAS-bsk^DN^;; rn-Gal4/UAS-bsk^DN^* 25°C (B) Blue: *w/+; ptc-Gal4, UAS-src.RFP/+; Sb/+* Red: *w/+; ptc-Gal4, UAS-src.RFP/+; Sb/UAS-GFP* (H, J) *w, UAS-bsk^DN^/w; ap-Gal4, UAS-src.RFP/+; UAS-bsk^DN^/+* 29°C (L) *w/+;; rn-Gal4/+* 29°C (M) *w/w, UAS-bsk^DN^;; rn-Gal4/UAS-bsk^DN^* 29°C (N) *w/+; UAS-bsk^AY^/+; rn-Gal4/+*

*Figure 2- figure supplement 2*: (A) *w/+; ptc-Gal4, UAS-src.RFP/+* 6 days AEL (B) *w/+; ptc-Gal4, UAS-src.RFP/+; UAS-egr/Sb* 6 days AEL (D) *w/+; ptc-Gal4, UAS-src.RFP/+* (E) *w/UAS-hid; ptc-Gal4,*

*UAS-src.RFP/+* (G) *w, UAS-bsk$^{DN}$/w; ptc-Gal4, UAS-src.RFP/+; UAS-egr/UAS-bsk$^{DN}$* (H) *w/+; ptc-Gal4, UAS-src.RFP/UAS-diap1; UAS-egr/Sb* (I) *w/w, UAS-p35; ptc-Gal4, UAS-src.RFP/+; UAS-egr/Sb*

*Figure 2—figure supplement 3*: (A, D) *w; ap-GAL4/UAS-src.RFP* (B) *w; ap-GAL4/UAS-src.RFP; UAS-EGFR$^{RNAi}$/+* (C, F) *w/w, UAS-bsk$^{DN}$; ap-GAL4/UAS-src.RFP; UAS-bsk$^{DN}$/+* (E) *w; ap-GAL4/UAS-src.RFP; UAS-dpp$^{RNAi}$/+* (J) *w/+;; rn-Gal4/+* (K) *w;; UAS-dpp$^{RNAi}$/rn-Gal4* (L) *w/w, UAS-bsk$^{DN}$;; rn-Gal4/UAS-bsk$^{DN}$*

*Figure 2—figure supplement 4*: (A) *w/yv; ptc-Gal4, UAS-src.RFP/+; UAS-EGFR$^{RNAi}$/+* (B) *w/yv; ptc-Gal4, UAS-src.RFP/+; UAS-dpp$^{RNAi}$/+*

*Figure 3*: (A) Blue: *w/+; ptc-Gal4, UAS-src.RFP/+* Red: *w/+; ptc-Gal4, UAS-src.RFP/+ UAS-jun$^{RNAi\#1}$/+* (C) Blue: *w/+; ptc-Gal4, UAS-src.RFP/+ 29°C* Red: *w/+; ptc-Gal4, UAS-src.RFP/+; UAS-jub$^{RNAi\#1}$/+ 29°C* (E) Blue: *w/+;; rn-GAL4/+ 29°C* Red: *w/w, UAS-bsk$^{DN}$; UAS-yki.GFP/+; rn-GAL4/UAS-bsk$^{DN}$ 29°C* (G) Blue: *w/+; ptc-Gal4, UAS-src.RFP/+ 29°C* Red: *w/w, UAS-bsk$^{DN}$; ptc-Gal4, UAS-src.RFP/UAS-yki.GFP; UAS-bsk$^{DN}$/+ 29°C* (I) Blue: *w/+; ptc-Gal4, UAS-src.RFP/+* Red: *w/+; ptc-Gal4, UAS-src.RFP/+; UAS-yki$^{RNAi\#1}$/+* (K) Blue: *w/+; ptc-Gal4, UAS-src.RFP/+ 29°C* Red: *w/+; ptc-Gal4, UAS-src.RFP/UAS-yki.GFP 29°C* (M) Blue: *w/+; ptc-Gal4, UAS-src.RFP/+; UAS-yki$^{RNAi\#1}$/+* Red: *w/UAS-bsk$^{DN}$; ptc-Gal4, UAS-src.RFP/+; UAS-yki$^{RNAi\#1}$/ UAS-bsk$^{DN}$* (O) Blue: *w/+; ptc-Gal4, UAS-src.RFP/UAS-yki.GFP 29°C* Red: *w/+; ptc-Gal4, UAS-src.RFP/UAS-yki.GFP; UAS-fj$^{RNAi}$/+ 29°C*

*Figure 3—figure supplement 1*: (A) *w/+; ap-Gal4, UAS-src.RFP/+; puc$^{E69}$/+* (B) *w/+; ap-Gal4, UAS-src.RFP/UAS-jun$^{RNA\#1i}$; puc$^{E69}$/+* (C) Blue: *w/+;; rn-Gal4/+* Red: *w/+; UAS-jun$^{RNAi\#1}$/+; rn-Gal4/+* (E) Blue: *w/+; ptc-Gal4, UAS-src.RFP/+* Red: *w/+; ptc-Gal4, UAS-src.RFP/UAS-jun$^{RNAi\#2}$* (G) Blue: *w/+;; rn-Gal4/+* Red: *w/+;; rn-Gal4/UAS-kay$^{RNAi}$* Green: *w/+; UAS-jun$^{RNAi}$/+; rn-Gal4/UAS-kay$^{RNAi}$*

*Figure 3—figure supplement 2*: (C) Blue: *w/+;; rn-Gal4/+* Red: *w/w, UAS-bsk$^{DN}$; UAS-bsk$^{DN}$/+; rn-Gal4/lats$^{e26-1}$* (E) *w; ptc-Gal4, UAS-src.RFP/+; UAS-yki$^{RNAi\#1}$/UAS-puc* (G) Blue: *w/+; ptc-Gal4, UAS-src.RFP/+ 29°C* Red: *w/+; ptc-Gal4, UAS-src.RFP/+; UAS-fj$^{RNAi}$/+ 29°C* (I) Blue: *w/+; ptc-Gal4, UAS-src.RFP/+* Red: *w/+; ptc-Gal4, UAS-src.RFP/UAS-fj; Sb/+*

*Figure 4*: (A) *w/+; ptc-Gal4, UAS-src.RFP/+* (B) *w/yv; ptc-Gal4, UAS-src.RFP/+; UAS-Ci$^{RNAi}$/+* (C) *w/+; ptc-Gal4, UAS-src.RFP/+; UAS-Ci$^{ACT}$/+* (D) Blue: *w/+; ptc-Gal4, UAS-src.RFP/+ 20°C* Red: *w/yv; ptc-Gal4, UAS-src.RFP/+; UAS-Ci$^{RNAi}$/+ 20°C* (E) Blue: *w/+; ptc-Gal4, UAS-src.RFP/+ 20°C* Red: *w/+; ptc-Gal4, UAS-src.RFP/+; UAS-Ci$^{ACT}$/+ 20°C* (F) Blue: *w/+; ptc-Gal4, UAS-src.RFP/+; UAS-Ci$^{ACT}$/+ 20°C* Red: *w/UAS-bsk$^{DN}$; ptc-Gal4, UAS-src.RFP/+; UAS-Ci$^{ACT}$/UAS-bsk$^{DN}$ 20°C*

*Figure 5*: (D) *w/+; ptc-Gal4, UAS-src.RFP/+ 29°C* (E) *w/+; ptc-Gal4, UAS-src.RFP/+; UAS-dTRAF1$^{RNAi\#1}$/+ 29°C* (F) Blue: *w/+;; rn-Gal4/+ 29°C* Red: *w/+;; UAS-dTRAF1$^{RNAi\#1}$/rn-Gal4 29°C* (G) Blue: *w/+; ptc-Gal4, UAS-src.RFP/+ 29°C* Red: *w/+; ptc-Gal4, UAS-src.RFP/+; UAS-dTRAF1$^{RNAi\#1}$/+ 29°C*

*Figure 5—figure supplement 1*: (E) *UAS-dTRAF1$^{RNAi\#2}$/Y; ptc-Gal4, UAS-src.RFP/+; Sb/+ 29°C* (F) Blue: *w/+; ptc-Gal4, UAS-src.RFP/+ 29°C* Red: *w/UAS-dTRAF1$^{RNAi\#2}$; ptc-Gal4, UAS-src.RFP/+ 29°C*

*Figure 5—figure supplement 2*: (A) Blue: *w/+; ptc-Gal4, UAS-src.RFP/+ 20°C* Red: *w/+; ptc-Gal4, UAS-src.RFP/+; UAS-Ci$^{ACT}$/UAS-dTRAF1$^{RNAi\#1}$ 20°C*

*Figure 6*: (A, G) *Canton-S* (B, D, H, J) *w; dll-Gal4, UAS-src.RFP/+* (C, I) *UAS-bsk$^{DN}$/Y; dll-Gal4, UAS-src.RFP/+; UAS-bsk$^{DN}$/+* (E, K) *w; dll-Gal4, UAS-src.RFP/+; UAS-egr/+*

# Acknowledgements

We thank the Vienna Drosophila RNAi Collection, the Transgenic RNAi Project, the Bloomington Drosophila Stock Center, and T Kornberg for fly stocks; R Nandez for technical assistance; R Harland for lab space, A Giraldez, V Greco, and L Cooley for helpful discussions. This work was supported in part by training grant T32 GM007499 to HRW and by award R01 CA069408-20 from the NIH/NCI to TXTX and PAB are Howard Hughes Medical Institute Investigators.

# Additional information

## Funding

| Funder | Grant reference number | Author |
|---|---|---|
| National Institutes of Health | R01CA069408-20 | Tian Xu |
| National Institutes of Health | Graduate Student Training Fellowship | Helen Rankin Willsey |
| Howard Hughes Medical Institute | HHMI Investigators | Philip A Beachy Tian Xu |
| National Institutes of Health | K99 | Xiaoyan Zheng Philip A Beachy |

The funders had no role in study design, data collection and interpretation, or the decision to submit the work for publication.

## Author contributions

HRW, Performed all experiments except the microarray, Conception and design, Acquisition of data, Analysis and interpretation of data, Drafting or revising the article; XZ, Performed the gene expression profiling and suggested its use in the analysis, Acquisition of data, Contributed unpublished essential data or reagents; JCP-P, TX, Conception and design, Analysis and interpretation of data, Drafting or revising the article; AJW, Identified differentially expressed genes, Analysis and interpretation of data, Drafting or revising the article; PAB, Performed the gene expression profiling and suggested its use in the analysis, Conception and design, Drafting or revising the article, Contributed unpublished essential data or reagents

## Author ORCIDs

José Carlos Pastor-Pareja, http://orcid.org/0000-0002-3823-4473
Tian Xu, http://orcid.org/0000-0002-2160-0027

# Additional files

## Supplementary files

• Supplementary file 1. Genes expressed in posterior (*hh+*) and/or *ptc* domain wing disc cells.

• Supplementary file 2. Differentially expressed genes between posterior (*hh+*) and *ptc* domain wing disc cells.

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
