## [Decision Letter]

Thank you for submitting your work entitled "Localized JNK Signaling Regulates Organ Size During Development" for consideration by *eLife*. Your article has been reviewed by three peer reviewers, one of whom is a member of our Board of Reviewing Editors and the evaluation has been overseen by the Reviewing Editor and Fiona Watt as the Senior Editor.

The reviewers have discussed the reviews with one another and the Reviewing editor has drafted this decision to help you prepare a revised submission.

Summary:

In this manuscript, Willsey et al. report a role for localized JNK signaling in regulating *Drosophila* organ size, particularly the developing wing. Mechanistically, Hh signaling from the posterior compartment activates JNK signaling in the A/P boundary by Ci-mediated transcriptional up-regulation of *dTRAF1*. The activated JNK signaling promotes cell proliferation and wing growth through Jun-independent, Jub-dependent Yki activation. While the connection between JNK and Yki has been previously reported, the findings of localized JNK signaling in organ size control and activation of JNK by Hh-Ci pathway are novel and important. However, there are concerns that should be addressed before this manuscript is considered for publication.

Essential revisions:

1) To verify the specificity of antibody, the authors used *rn-Gal4* and *ptc-Gal4* to knock down *bsk* (Figure 1, Figure 1—figure supplement 1). Actually *ap-Gal4* should be used in this situation, as it is expressed only in the dorsal compartment, and the ventral part could serve as an internal control. Similarly, *ap-Gal4* should be used to knock down *hep*, since some p-JNK staining still presents in the *hep* mutant clone (Figure 1').

2) A control UAS line (e.g. UAS-LacZ or *UAS -GFP*) should be included to exclude the possibility that expression of any protein by *ptc*- or *rn-Gal4* may disturb wing development and affect wing size.

3) Is there any effect on wing disc size by blocking JNK activity?

4) Does *ptc*>*egr*-induced wing disc enlargement depends on Bsk?

5) Does *ptc*>*egr* increases the adult wing size?

6) To show the increased *ptc*>*egr* wing disc size is not a consequence of apoptosis, authors should block apoptosis by expressing p35.

7) A positive control should be included to validate CCP3 staining (Figure 2—figure supplement 1).

8) There is no evidence that the endogenous JNK signaling regulates cell proliferation. The authors should check cell proliferation (Edu staining) in *rn*- or *ptc*>*bsk^DN^* discs. The non-cell autonomous increase of cell proliferation in *ptc*>*egr* discs could be triggered by caspase activation, rather than a direct outcome of JNK activation. To discriminate the two possibilities, *diap1* should be added to block caspase activation.

9) Though Jun is not required by JNK to regulate wing size, what about Fos?

10) The effect of *UAS-Yki* on *UAS-bsk^DN^* could be additive effect, but not rescue. The authors should check whether *ptc*> or *rn*>*bsk^DN^*-induced small wing phenotype could be suppressed in heterozygous *lats* mutants.

11) *UAS-bsk^DN^* is probably not strong enough to enhance the *ptc*>*yki^RNAi^*phenotype, what about *UAS-puc*?

12) Though *fj* appears to be involved in ectopic Yki-triggered wing growth, is it required by endogenous *bsk* and *yki* to regulate wing growth? How do the authors explain this? Is *fj* acting via control of Ft or Ds in this process? To prove this point more clearly, the authors should overexpress *fj* with PtcGal4. Also, does the *fj-lacZ* reporter show a stripe pattern similar to pJNK in the 3rd instar larval wing disc?

13) Does *dTRAF1^RNAi^* block CiACT-triggered wing growth?

14) Does expression of *dTRAF1* increase wing size in a Bsk-dependent manner?

15) Surprisingly, knockdown of Jun did not affect wing size-the authors invoke a non-canonical pathway, but could the knockdown have been incomplete? Also, is there redundancy with Kayak? Similarly, the argument that this signaling is non-canonical is based on the *puc-lacZ* reporter- could this instead be due to delayed reporter activity, or reduced sensitivity?

[Editors' note: further revisions were requested prior to acceptance, as described below.]

Thank you for resubmitting your work entitled "Localized JNK Signaling Regulates Organ Size During Development" for further consideration at *eLife*. Your revised article has been favorably evaluated by Fiona Watt (Senior editor), a Reviewing editor, and two reviewers, one of whom is a member of our Board of Reviewing Editors. The manuscript has been improved but there are some remaining issues that need to be addressed before acceptance, as outlined below:

The authors have addressed all the concerns well, except two below.

1) Does expression of *dTRAF1* increase wing size in a Bsk-dependent manner?

The authors responded that ptc>*dTRAF1* is lethal. What about other *Gal4* drivers, e.g. *rn-Gal4, ap-Gal4*? What Cha et al. showed is that *dTRAF1*-induced cell death depends on JNK. It remains unknown whether *dTRAF1* regulates JNK-dependent cell proliferation and growth. The question is quite crucial for this manuscript.

2) Redundancy of Kayak. Can the authors provide evidence of the efficacy of their Kayak knockdown. The provided experiment showing that double knockdown of Kayak and Jun has no effect on growth is only worthwhile if the knockdowns are effective.

---

## [Author Response]

1) To verify the specificity of antibody, the authors used rn-Gal4 and ptc-Gal4 to knock down bsk (Figure 1, Figure 1—figure supplement 1). Actually ap-Gal4 should be used in this situation, as it is expressed only in the dorsal compartment, and the ventral part could serve as an internal control. Similarly, ap-Gal4 should be used to knock down hep, since some p-JNK staining still presents in the hep mutant clone (Figure 1').

We thank the reviewers for this suggestion. We now use *ap-Gal4* to inhibit JNK signaling in the dorsal compartment and see a specific reduction in pJNK in those cells (*ap*>*puc*). This data is now included in the text and figures (paragraph one, subheading “JNK is active in the developing Drosophila wing pouch”; Figure 1—figure supplement 1). *hep^r75^* (used in the clonal analysis) is likely not a null allele since hemizygous mutant embryos complete dorsal closure, which likely explains the minor residual pJNK staining in the mutant clones. Unfortunately none of the available *UAS-hep^RNAi^* lines are strong enough to induce the known *hep* mutant phenotype of a split thorax, nor are they strong enough to abolish pJNK staining (data not shown). So unfortunately we cannot perform this experiment precisely as requested. However, we note that we have done 10 independent experiments to validate the specificity of the antibody and are confident in its fidelity.

2) A control UAS line (e.g. UAS-LacZ or UAS -GFP) should be included to exclude the possibility that expression of any protein by ptc- or rn-Gal4 may disturb wing development and affect wing size.

We now show that expression of *UAS-GFP* by *ptc-GAL4* does not affect wing size (*ptc*>*GFP*). We have now included the new data in the text and in the figures (paragraph one, subsection “Localized JNK activity regulates global wing size”; Figure 2—figure supplement 1).

3) Is there any effect on wing disc size by blocking JNK activity?

We now show that blocking JNK causes a reduction in wing disc size (*ap*>*bsk^DN^*). We have included the new data in the text and in the figures (paragraph one, aforementioned subsection; Figure 2—figure supplement 1).

4) Does ptc>egr-induced wing disc enlargement depends on Bsk?

We now show that *ptc*>*egr*-induced disc enlargement depends on *bsk (ptc*>*egr, bsk^DN^*). In fact, these discs are significantly smaller than even control discs (p= 0.0078). We have now included the new data in the text and in the figures (paragraph two, aforementioned subsection; Figure 2, disc image in Figure 2—figure supplement 2).

5) Does ptc>egr increases the adult wing size?

We tried to assess whether *ptc*>*egr* causes an increase in adult wing size, but these animals were larval lethal. The overgrowth of the disc likely precludes proper pupation. We have now added this point to the text (paragraph two, aforementioned subsection).

6) To show the increased ptc>egr wing disc size is not a consequence of apoptosis, authors should block apoptosis by expressing p35.

We now show that expression of *UAS-p35* with *UAS-egr* does not abolish the size effect of *UAS-egr (ptc*>*egr, p35*). We have now included the new data in the text and in the figures (paragraph two, aforementioned subsection; Figure 2, disc image in Figure 2—figure supplement 2).

7) A positive control should be included to validate CCP3 staining (Figure 2—figure supplement 1).

We now show that expression of *UAS-bsk^AY^*, a constitutively active JNK allele, induces CCP3 staining in the wing. We have now included the new data in the figures (Figure 2—figure supplement 1).

8) There is no evidence that the endogenous JNK signaling regulates cell proliferation. The authors should check cell proliferation (Edu staining) in rn- or ptc>bsk^DN^ discs. The non-cell autonomous increase of cell proliferation in ptc>egr discs could be triggered by caspase activation, rather than a direct outcome of JNK activation. To discriminate the two possibilities, diap1 should be added to block caspase activation.

We now show that inhibiting JNK signaling causes a reduction in proliferation by phosphorylated histone 3 staining (*ap*>*bsk^DN^*). We have included this new data in the text and in the figures (paragraph four, aforementioned subsection; Figure 2—figure supplement 1). We now also show that expression of *UAS-diap1* does not block the growth effect of *UAS-egr (ptc*>*egr, diap1*). We have included this new data in the text and in the figures (paragraph two, same subsection; Figure 2, disc image in Figure 2—figure supplement 2).

9) Though Jun is not required by JNK to regulate wing size, what about Fos?

We now show that inhibiting Fos does not alter wing size (*rn*>*kay^RNAi#1,2^*). We have now included the new data in the text and in the figures (paragraph one, subsection “Non-canonical JNK signaling regulates size”; Figure 3—figure supplement 1).

10) The effect of UAS-Yki on UAS-bsk^DN^ could be additive effect, but not rescue. The authors should check whether ptc> or rn>bsk^DN^-induced small wing phenotype could be suppressed in heterozygous lats mutants.

We now show that the *rn*>*bsk^DN^* wing phenotype can be partially suppressed in a heterozygous *lats* mutant background (*rn*>*bsk^DN^; lats^e2b-1/+^*). We have now included the new data in the text and in the figures (paragraph two, same subsection; Figure 3—figure supplement 2).

11) UAS-bsk^DN^ is probably not strong enough to enhance the ptc>yki^RNAi^ phenotype, what about UAS-puc?

We now show that *UAS-puc* does not enhance the *ptc*>*yki^RNAi^*phenotype (*ptc*>*yki^RNAi^, puc*). We have now included the new data in the text and in the figures (paragraph three, same subsection; Figure 3—figure supplement 2).

12) Though fj appears to be involved in ectopic Yki-triggered wing growth, is it required by endogenous bsk and yki to regulate wing growth? How do the authors explain this? Is fj acting via control of Ft or Ds in this process? To prove this point more clearly, the authors should overexpress fj with PtcGal4. Also, does the fj-lacZ reporter show a stripe pattern similar to pJNK in the 3rd instar larval wing disc?

We have now overexpressed *fj* and found it also reduces wing size. Therefore, we cannot simply conclude that *fj* is required by endogenous Bsk and/or Yki to regulate growth. We have now made this clear in the text and figures (paragraph four, same subsection; Figure 3—figure supplement 2).

*fj-lacZ* is known to be present in a gradient in the wing disc, highest at the A/P and D/V boundaries, emanating distally (Villano and Katz, 1995). Overall, signaling downstream of Yki is intricate and has not been worked out.

13) Does dTRAF1^RNAi^ block CiACT-triggered wing growth?

We now show that *UAS-dTRAF^RNAi^* can modulate Ci^ACT^-triggered wing growth (*ptc*>Ci^ACT^, *dTRAF^RNAi^*), further strengthening our finding. We have now included the new data in the text and in the figures (paragraph three, subsection “Hh sets up pJNK by elevating *dTRAF1* expression”; Figure 5—figure supplement 2).

14) Does expression of dTRAF1 increase wing size in a Bsk-dependent manner?

We tried to determine whether expression of *UAS-dTRAF1* in the *ptc* domain increases wing size in a Bsk-dependent manner, but unfortunately expressing *UAS-dTRAF1* is lethal. Nevertheless, it has been shown that *dTRAF1* function in the eye is Bsk-dependent (Cha et al., 2003). We have now included the new data and discussed this in the text (p. 11-2, para. 2, line 257-259).

15) Surprisingly, knockdown of Jun did not affect wing size-the authors invoke a non-canonical pathway, but could the knockdown have been incomplete? Also, is there redundancy with Kayak? Similarly, the argument that this signaling is non-canonical is based on the puc-lacZ reporter- could this instead be due to delayed reporter activity, or reduced sensitivity?

Null mutant clones of *jun* do not show a phenotype in the wing (Kockel et al., 1997). Furthermore, *puc-lacZ* is both a sensitive and quick JNK signaling reporter, as indicated by its fast and robust response to JNK activation (McEwen and Peifer, 2005). We note that *UAS-jun^RNAi^* is strong enough to show an effect on *puc-lacZ* expression in the stalk region of the wing (Figure 3—figure supplement 1). We now show that inhibition of *kayak* does not have an effect on wing size (*rn*>*kay^RNAi^*), and is not redundant with Jun (*rn*>*jun^RNAi^, kay^RNAi^*). These data are consistent with previous reports that *jun/fos* do not control wing growth (Kockel et al., 1997). We have now included the new data and discussed this in the text (Paragraph one, subsection “Non-canonical JNK signaling regulates size”; Figure 3—figure supplement 1).

*[Editors' note: further revisions were requested prior to acceptance, as described below.] 1) Does expression of dTRAF1 increase wing size in a Bsk-dependent manner?*

*The authors responded that ptc>dTRAF1 is lethal. What about other Gal4 drivers, e.g. rn-Gal4, ap-Gal4? What Cha et al. showed is that dTRAF1-induced cell death depends on JNK. It remains unknown whether dTRAF1 regulates JNK-dependent cell proliferation and growth. The question is quite crucial for this manuscript.*

The other *Gal4* drivers mentioned (*rn-Gal4* and *ap-Gal4*) express *Gal4* in many more cells than *ptc-Gal4*, so over-expression of *dTRAF1* will certainly be lethal. We show that *dTRAF1* is required for growth and cell proliferation, as inhibition of *dTRAF1* in the wing leads to a small wing phenotype and a loss of pJNK staining (Figure 5). Indeed, *dTRAF1* null mutants fail to grow during the larval stages and have very small imaginal discs (Cha et al., Figure 5). This loss of function experiments show that in addition to cell death, *dTRAF1* is involved in regulation of growth.

*2) Redundancy of Kayak. Can the authors provide evidence of the efficacy of their Kayak knockdown. The provided experiment showing that double knockdown of Kayak and Jun has no effect on growth is only worthwhile if the knockdowns are effective.*

These kayak RNAi lines induced the typical JNK-phenotype, thorax closure defect, when driven by *ap-Gal4*, confirming their efficacy. We have added this in the legend of Figure 3—figure supplement 1.